# Hardware-Algorithm Co-Design for Hyperdimensional Computing Based on Memristive System-on-Chip

**Yi Huang, Alireza Jaberi Rad, Qiangfei Xia**
Department of Electrical and Computer Engineering
University of Massachusetts Amherst
Amherst, MA 01003
qxia@umass.edu

## Abstract

Hyperdimensional computing (HDC), utilizing a parallel computing paradigm and efficient learning algorithm, is well-suited for resource-constrained artificial intelligence (AI) applications, such as in edge devices. In-memory computing (IMC) systems based on memristive devices complement this by offering energy-efficient hardware solutions. To harness the advantages of both memristive IMC hardware and HDC algorithms, we propose a hardware-algorithm co-design approach for implementing HDC on a memristive System-on-Chip (SoC). On the hardware side, we utilize the inherent randomness of memristive crossbar arrays for encoding and employ analog IMC for classification. At the algorithm level, we develop hardware-aware encoding techniques that map data features into hyperdimensional vectors, optimizing the classification process within the memristive SoC. Experimental results in hardware demonstrate 90.71% accuracy in the language classification task, highlighting the potential of our approach for achieving energy-efficient AI deployments on edge devices.

## 1 Introduction

As artificial intelligence (AI) models continue to grow in complexity and scale, the energy consumption required for these models has been increasing dramatically Patterson et al. [2021]. The surge in energy demand has highlighted the energy efficiency in developing future AI systems, especially in scenarios where resources are limited, such as edge applications. Therefore, advancements at both hardware and algorithm levels are highly demanded for deploying AI models across a diverse range of applications and devices.

At the hardware level, in-memory computing (IMC) hardware based on memristor devices offers a promising solution by enabling computing within where data is stored Huang et al. [2024]. This approach reduces the time and energy associated with data transfer between memory and processing units, a bottleneck in von Neumann architectures. Memristive IMC hardware leverages the inherent properties of memristor devices to implement parallel vector-matrix multiplication (VMM) by using physical laws, accelerating the inference of neural networks Li et al. [2018], Wan et al. [2022], Wen et al. [2024]. Advancements at the device level, such as the increased number of conductance states of memristor devices Rao et al. [2023], combined with progress at the circuit level, such as the integration of multiple memristor crossbar arrays into a single chip Gallo et al. [2023], Zhang et al. [2023], have enhanced the overall capability of IMC systems. In parallel, hardware-algorithm co-designs for memristive crossbar arrays, aimed at achieving arbitrary precision in weight matrix, have enabled high-precision IMC Song et al. [2024].

At the algorithm level, hyperdimensional computing (HDC), inspired by biological brains, provides an energy-efficient and hardware-friendly approach by using hyperdimensional vector (HV) repre-

38th Second Workshop on Machine Learning with New Compute Paradigms at NeurIPS 2024(MLNCP 2024).

sentations of data Kanerva [1988], Chang et al. [2023]. In HDC, all computations are executed in high-dimensional space, facilitating fast training and parallel inference for the classification tasks at the edge. HDC typically includes two stages: the encoding stage, where original data is transformed into HVs that capture the key features, and the inference stage, where encoded HVs are compared with pre-trained HVs to produce final classification results. Various encoding methods, including low-power sparse encoding Imani et al. [2017a] and encoding based on Nyström method Zhao et al. [2023], have been explored to achieve efficient and diverse data representations. For the inference of HDC, adaptive training methods have been proposed for robust and efficient inference Hernandez-Cano et al. [2021]. Beyond the conventional associative memory widely employed as classifiers, neural networks have also been used to enhance the voice recognition accuracy Imani et al. [2017b].

To fully unleash the energy efficiency of IMC hardware and HDC algorithm, existing research has explored the implementation of HDC using IMC hardware based on different memristive devices. Early studies on resistive random-access memory (RRAM)-based IMC hardware for HDC focused on three-dimensional integration of memristor devices to increase device density, facilitating the storage and computation of HVs Li et al. [2016], Wu et al. [2018]. These works adhere to conventional HDC computing paradigm but improve the speed and energy efficiency of HDC by exploiting the parallelism inherent in IMC hardware and HDC algorithms. With the development of memristor devices, encoding and classification based on phase change memory (PCM) Karunaratne et al. [2020] and ferroelectric FET (FeFET) Huang et al. [2023] have been developed with tailored peripheral circuits to further enhance energy efficiency of HDC. Meanwhile, as the computing capability of memeristive hardware increases, there is a growing need in hardware-algorithm co-designs for HDC to fully harness the potential of IMC hardware. The results shown in Iwasaki and Shintani [2023], and Thomann et al. [2023] highlight the co-design approach to balance energy efficiency and classification accuracy. However, existing works primarily focus on utilizing binary memristor devices to implement HDC with digital IMC. Moreover, most HDC designs for IMC hardware have been validated primarily through simulations based on memristor models. While these simulations provide valuable insights, there is a lack of studies that demonstrate these designs using experimental hardware. Additionally, the energy efficiency of HDC algorithms using IMC hardware is compromised by the need for off-chip peripherals. These off-chip peripherals are required for data transfer and the processing of intermediate data, which increases the data transfer latency and energy consumption.

To address these issues and utilize IMC for HDC, we propose a hardware-algorithm co-design approach to improve HDC algorithm for our memristive System on Chip (SoC), which integrates ten memristive crossbar arrays to perform analog VMM. The major contributions of our work are:

1. Utilizing the inherent randomness of memristor devices to map data features to HVs with a single-step VMM.

2. Using multiple conductance states as weights for a single-layer perceptron to fully leverage the benefits of analog IMC for energy-efficient classification.

3. Implementing both encoding and the single-layer memristive perceptron in hardware by coordinating multiple memristive computing cores within one SoC and on-chip peripheral circuits, achieving experimental accuracy of 90.71 % in language classification.

## 2 Hardware and Algorithm Co-Design for HDC with IMC

### 2.1 Memristive SoC and Analog VMM

The IMC hardware used in this work is an evaluation kit with a memristive SoC. TetraMem [2024] As shown in Figure 1, the SoC includes ten computing cores, each integrating one memristive crossbar array and peripheral circuits, such as digital-to-analog converters (DACs) and analog-to-digital converters (ADCs). Each computing core contains a $248 \times 256$ one-transistor-one-memristor (1T1R) crossbar array, which is used to perform analog VMM. In addition to the analog computing cores, the SoC integrates a RISC-V CPU that facilitates data transfers between multiple computing cores and manages peripheral operations. It also includes digital circuits such as on-chip memory for intermediate data storage and interfaces for on-chip communications.

To implement the analog VMM, input vectors are converted to voltages by on-chip DACs and applied to the rows of the memristive crossbar arrays. These input voltages are then multiplied by the matrix values, which are represented by the conductance of memristor devices. The output vectors are

generated from the accumulated currents collected from the columns of the memristive crossbar array. On-chip transimpedance amplifiers (TIAs) convert the output currents to voltages, which are then digitized by ADCs for further processing.

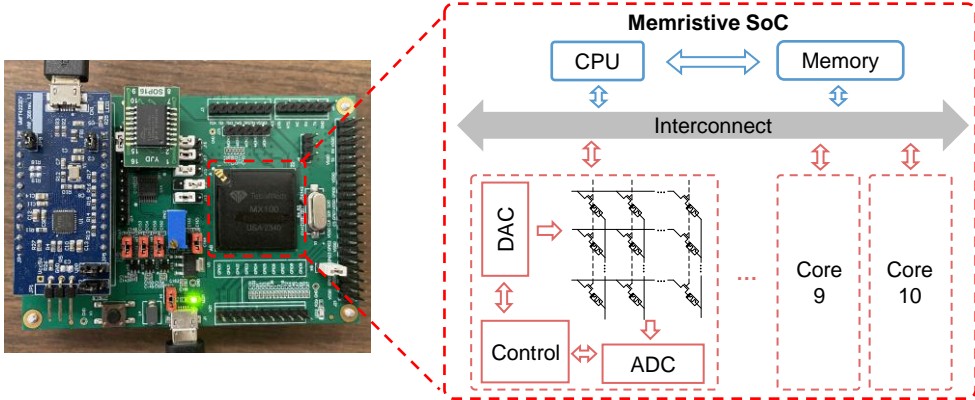

Figure 1: The photograph of MX100 evaluation kit TetraMem [2024] with memristive SoC and the SoC architecture.

## 2.2 Hardware-Aware Encoding and Memristive Classifier

From the computing perspective, conventional HDC using fully digital processing is not well-suited to fully leverage the parallel analog VMM capabilities of multiple computing cores. To take advantage of the energy efficiency of the memristive SoC, it is necessary to develop analog hardware-friendly encoding methods and classifiers for HDC. Furthermore, with limited hardware resources, specifically, the ten $248 \times 256$ cores for both encoding and classifiers, it is challenging to use HVs with 10000 dimensions typically employed in conventional digital HDC Hernandez-Cano et al. [2021], Rahimi et al. [2016]. Therefore, it is essential to balance energy efficiency and classification accuracy through HDC algorithm designs. To address these challenges, we propose VMM-based encoding, hardware-aware processing, and a single-layer memristive perceptron as the classifier, unleashing the parallelism and efficiency of IMC and HDC.

The workflow of conventional HDC and our co-design approach for a language classification task is illustrated in Figure 2. Initially, the letters in sentences from different languages are mapped to feature vectors, which are represented by voltages applied to the rows of memristive crossbar arrays in the hardware implementation. These feature vectors are then transformed into HVs with a single-step VMM utilizing the inherent randomness of memristive crossbar arrays as random matrices. After mapping the feature vectors to HVs, post-processing steps are performed to generate a set of HVs for each language, which are used for the training and inference of the classifier. To address the imperfections of analog VMM caused by hardware non-idealities, we apply the hardware-aware processing to the HVs before feeding them to the classifier. In conventional HDC, associative memory is used as the classifier to achieve high energy efficiency at the cost of low accuracy. Since implementing associative memory and single-layer perceptrons with analog VMM requires the same amount of hardware resources, we opt for a single-layer perceptron implemented in memristive crossbar arrays for classification, as it improves the classification accuracyImani et al. [2017b].

### 2.2.1 VMM-Based Transformation

The first step in encoding for HDC is the transformation of data features into HVs. Unlike conventional HDC, which relies on random basis HVs generated in software and stored in memory, we utilize the inherent randomness of memristor devices to create a random matrix for the transformation. By employing the hardware-based random matrix, the transformation can be efficiently implemented using a single-step VMM on memristive crossbar arrays.

In the language classification task, we first map each letter in text samples to a binary vector, where '1' is represented by the high voltage and '0' by the low voltage. Each character feature is represented by a 27-dimensional vector for the 27 possible characters (including a whitespace character). We

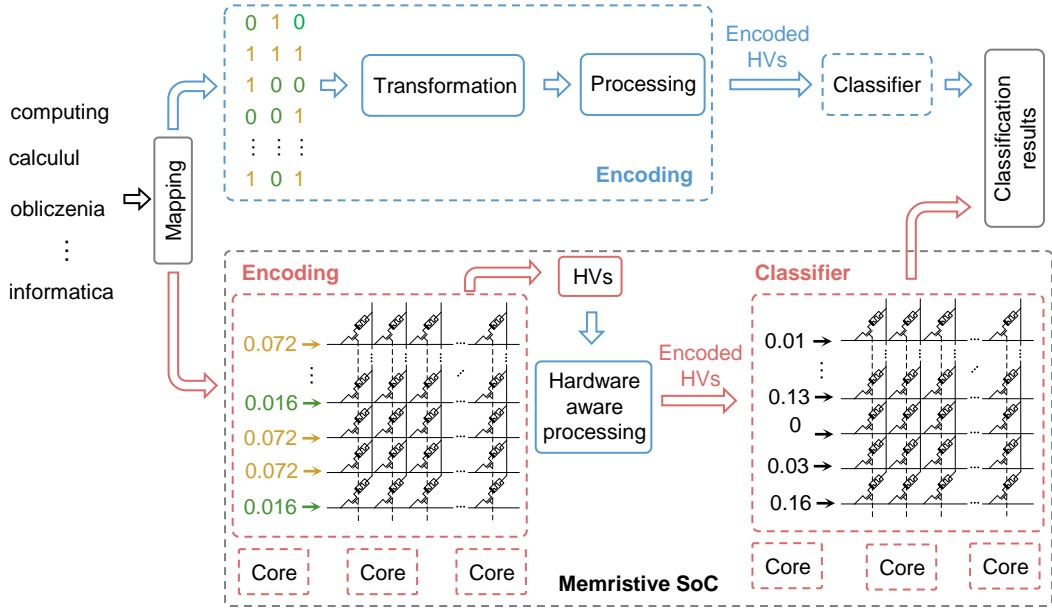

Figure 2: Workflow of conventional HDC (top flow) and the hardware-algorithm co-design for HDC based on Memristive SoC (bottom flow).

then combine every three consecutive letters to create 'trigram' vectors as trigram vectors balance the accuracy and energy consumption Rahimi et al. [2016]. These trigram vectors serve as feature vectors for texts from different languages. An analog VMM with the 81-dimensional trigram vectors as inputs is used to transform these feature vectors into HVs. The random matrix in the analog VMM is represented by the random conductance of memristor devices in the crossbar array, which is generated by applying SET voltages to the arrays. Considering the number of on-chip computing cores, we utilize $81 \times 128$ subarrays from 4 memristive crossbar arrays to perform the VMM-based transformation, resulting in 512-dimensional HVs presented by currents to encode the features of every three consecutive letters. After transforming all trigram vectors in a text sample to HVs, we use the least significant bit of each ADC output to determine the sign of the corresponding entry of the HVs associated with each trigram vector. As a result, the HVs for each trigram vector produced by the VMM-based transformation are a set of HVs with values of either -1 or 1, which are used to generate the input HVs for the classifier.

### 2.2.2 Noise-Tolerant Processing

Different from encoding implemented in software, encoding based on analog hardware is subject to noise due to the non-idealities of memristor devices and peripheral circuits, such as TIAs and ADCs. In the VMM-based transformation, output currents fluctuate within a small range, leading to shifts in the ADC results we used to determine the HVs. Since the least significant bits of ADC results are highly sensitive to the fluctuations of output currents, the HVs from the ADC results propagate the noise to the classifiers, impacting classification accuracy. However, upon analyzing experimental VMM results from the memristive SoC, we observe that most outputs fluctuate within a small range of $\pm 2.77mV$, which only impacts the lower few bits of ADC results for the 8-bit on-chip ADCs. To mitigate the impact of these fluctuations on outputs from identical inputs, we choose another bit position of the ADC results to determine the sign of the corresponding entry in the HVs by analyzing the noise introduced by the memristive crossbar arrays used for encoding. For instance, using the third least significant bit of the ADC results ensures this bit remains consistent when the fluctuations only affect the least two bits of the ADC results. This approach can tolerate noise from the non-idealities of analog VMM on the memristive SoC while effectively distinguishing feature vectors from text samples in different languages.

After determining the sign of the entries of the HVs, all HVs from one text sample are summed to generate an encoded HV that represents the features of the text sample. Since the HVs from each

trigram vector contain both -1 and 1, the resulting encoded HVs range from negative to positive values. In conventional HDC, these encoded HVs are binarized before being used as inputs to the classifier, as binary representation is more friendly to digital computing systems. In contrast, our memristive SoC allows for multilevel voltage inputs to the classifier because there are 8-bit on-chip DACs connected to each row of the memristive crossbar arrays. Therefore, instead of binarizing the encoded HVs, we quantize their values to multi-bit for the classifier. These multilevel HVs preserve more feature information from each text sample, which compensates the relatively low dimensions of HVs used due to the limitation of hardware resources (512 dimensions instead of the typical 10000 dimensions). The noise-tolerant processing to generate the encoded HVs can be implemented in the on-chip CPU illustrated in Figure 2.

### 2.2.3 Single-Layer Memristive Perceptron for Classification

After encoding the text data features into HVs, language classification becomes a relatively simple task. In conventional HDC, associative memory is commonly used as the classifier due to its efficiency. However, a neural network trained with the stochastic gradient descent (SGD) algorithm can achieve higher classification accuracy, as demonstrated by Imani et al. [2017b]. Conventional HDC prioritizes energy efficiency over accuracy because neural networks typically consume more energy than associative memory. However, our approach utilizes memristive crossbar arrays to implement a single-layer perceptron in a single step using parallel analog VMM. This implementation leverages the same computational demand as associative memory and enables us to benefit from both the energy efficiency of the memristive hardware and the improved accuracy of a neural network-based classifier.

In the language classification task, we use a perceptron with 512 input neurons and 21 output neurons as the classifier, corresponding to the 21 European languages to be classified. Instead of binary voltages used in the VMM-based transformation, the encoded HVs from the hardware-aware encoding are represented by analog voltages ranging from 0 - 0.141 V. These analog voltages serve as the inputs to the single-layer perceptron. The synaptic weights of the perceptron are initially trained offline and then mapped to the conductance values of the memristor devices. The weight conductance is programmed to the memristive crossbar arrays within the computing cores by applying SET/RESET voltages to the memristor devices before the inference. Since each core can accommodate a maximum of 248 rows, less than the 512 input neurons, we distribute the weight conductance across multiple computing cores. During the inference, the encoded HVs are applied to the rows of the memristive crossbar arrays, and the maximum output current from the columns indicates the classification results.

## 3 Experimental Results

The dataset for the language classification task consists of short text samples from 21 European languages, each includes 1000 text samples that are all transliterated into the Latin alphabet Rahimi et al. [2016]. We use 70 % of these samples for training while the remaining 30 % for evaluation. To evaluate the HDC algorithm design for our memristive SoC, we first simulate the encoding and classifier in software using the hardware-aware encoding method and single-layer perceptron. The average classification accuracy achieved is 96.71 %, as shown in Figure 3. This result demonstrates that our proposed hardware-aware encoding, combined with the single-layer perceptron, achieves a classification accuracy comparable to the 96.7 % reported in previous work Rahimi et al. [2016]. But the 512 dimension of HVs used for encoding in our approach is much lower than 10000 dimensions typically used in conventional HDC, significantly saving both computing resources and energy consumption.

After confirming the accuracy achieved by our proposed co-design method through simulations, we implement both the encoding and classifier separately on our memristive SoC to study the impact of non-idealites in memristor devices and peripheral circuits on classification accuracy. For the configuration with hardware encoding and software classifier, feature vectors are encoded using analog VMM by coordinating four computing cores within the SoC, while the training and inference of the single-layer perceptron are implemented in software. In contrast, for the setup with software encoding and hardware classifier, the encoding is simulated and the single-layer perceptron is trained offline on a host PC. The trained weights are then programmed to three memristive subarrays within the SoC for hardware-based classification. For the fully hardware-based inference, encoded HVs are generated from the encoding implemented on four computing cores within the memristive SoC, while

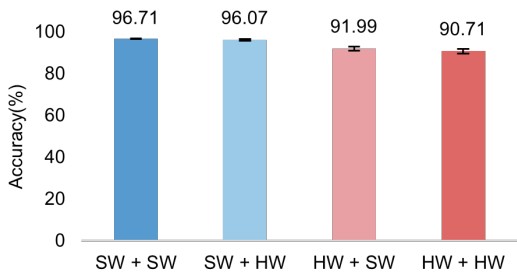

Figure 3: Classification accuracies of the 21 European languages with different setups, encoding + classifier implementations. (SW: Software implementation, HW: Hardware implementation)

the hardware-aware encoding and single-layer perceptron are distributed across three computing cores and the on-chip CPU within the SoC.

We compare the results, shown in Figure 3, from pure software simulations, hardware implementations and these mixed setups. The configuration with a hardware-based classifier and software-based encoding achieves higher accuracy, closely matching the results of the pure software simulation, compared to the setup with hardware-based encoding and a software-based classifier. These results indicate that the memristive perceptron classifier is robust against noise introduced by hardware, whereas the encoding process is more sensitive to hardware non-idealities, which negatively impacts classification accuracy. The fully hardware implementation achieved an average classification accuracy of 90.71%, with the classification accuracy for each language shown in Figure 4. The experimental results, showing comparable accuracies across most languages, demonstrate the effectiveness of our hardware-algorithm co-design with limited hardware resources. The dimensionality (512) of HVs employed in our approach is significantly lower than the typical dimensionality (10000) used in conventional HDC Rahimi et al. [2016], Imani et al. [2017b], Iwasaki and Shintani [2023]. This reduction translates to 94.8 % savings in hardware requirements for both encoding and classification.

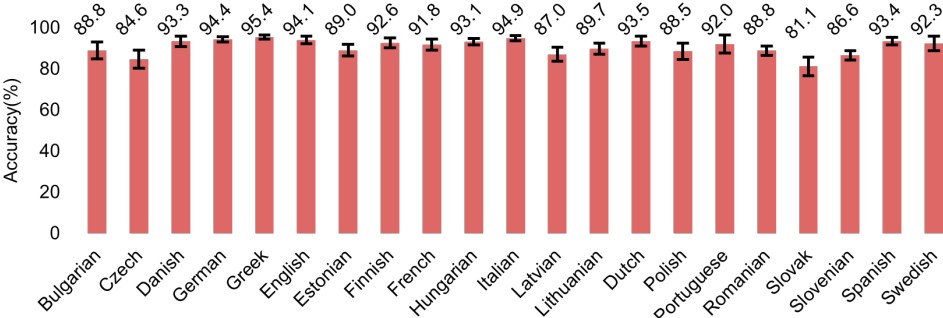

Figure 4: Classification accuracy for each of the 21 European languages with hardware-implemented encoding and classifier.

## 4 Conclusion

In conclusion, we propose a hardware-algorithm co-design approach to leverage IMC and HDC within a memristive SoC. By coordinating multiple memristive computing cores within the SoC to implement hardware-aware encoding and a single-layer perceptron, we demonstrate the effectiveness of the proposed co-design with a language classification task. The simulation results, achieving an average classification accuracy of 96.71% validate our modifications to the HDC algorithm, while the experimental results with an average classification accuracy of 90.71 % confirm the feasibility of our hardware implementations. The hardware-algorithm co-design paves the way to harness IMC for energy-efficient HDC in edge applications. Future work will focus on increasing the dimensions of HVs as hardware resource allows and applying this approach to other more challenging tasks.

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
