# OpenReview forum: "Hardware-Algorithm Co-Design for Hyperdimensional Computing Based on Memristive System-on-Chip"
_NeurIPS.cc/2024/Workshop/MLNCP — MLNCP Poster_

### Official Review · Reviewer_Wyir · 2024-09-23
**HDC Hardware-Algorithm co-design use Analog IMC for simple language processing task**

**Rating:** 6
**Confidence:** 3

**Review:**

This paper proposed an simple HDC  Hardware-Algorithm co-design based on analog IMC for on chip computing. Co-design is an interesting topic with practical benefits, especially for HDC-based methods which have desirable properties for hardware implementation.

Strong points:
- Analog implementation of HDC applications seems to be an promising direction. Although authors should definitely consider adding more motivation for analog vs. digital design in the future.
- Although a bit un-organized, authors provided many details regarding analog SoC implementation which is important.

Weak points:
- Writing and structure of the paper should be improved significantly. At its current stage it is difficult to follow some of the design decisions are not well motivated (i.e. why trigram).
- The capability of proposed system seems to be very limited, it is only suitable for tasks with a small number of symbols (i.e. the European language dataset demonstrated in the paper) and low HDC dimensionality. How would it performs in other more challenge tasks.
-  Although the authors claims algorithm co-design, there is little novelty regarding proposed algorithms as N-gram based encoding has been extensive studies in the context of HDC even with simple binary HDC architecture.
- Following previous point on novelty, it is unclear what is new being offered here by using a perceptron instead of traditional HDC retraining steps (which is also perceptron algorithm by considering class hypervectors the weights and adjusting them iteratively). Furthermore, why does this part needs pretraining and is it an realistic assumption.
- Hardware-Algorithm co-design is an popular research field for HDC but this paper does not include any comparisons to other design.
- Evaluation section failed to show the benefits of proposed design (efficiency? time?), and accuracy degradation seem quite significant (96->90).

Overall, my main concern for this paper is the lack of novelty and the lack of valid evaluations and comparisons that can demonstrate the practical benefits of the design.

---

### Official Review · Reviewer_ELaM · 2024-10-02
**Hardware-Algorithm Co-Design for HDC**

**Rating:** 7
**Confidence:** 3

**Review:**

The authors devleop a HDC application using a SoC from TetraMem.

While I don't agree with the authors' statement that there have not been previous analog / memristive demonstrations of HDC, I think it is commenable to have mapped a simple HDC problem onto a SoC containing Memristors and demonstrated an application on the hardware while keeping the task performance relatively high.

The principle of using intrinsic noise of memristor programming to generate random basis' is also novel and this claim should be tempered since it appears as one of the main contributions of this paper.

I did not understand the phrase "Taking advantage of on-chip peripheral circuits to mitigate the impact of noise from hardware non-idealities on classification accuracy" because I did not see in the paper how peripheral circuits of the SoC achieved this in the paper. What I understoof was that you didn't use the LSbits of the ADC since they were subject to noise, but I wouldn't consider this taking advantage ?

---

### Decision · Program_Chairs · 2024-10-10

Accept (Poster)